# Synthesis of Organic Iodine Compounds in Sweetcorn under the Influence of Exogenous Foliar Application of Iodine and Vanadium

**DOI:** 10.3390/molecules27061822

**Published:** 2022-03-11

**Authors:** Marlena Grzanka, Sylwester Smoleń, Łukasz Skoczylas, Dominik Grzanka

**Affiliations:** 1Department of Plant Biology and Biotechnology, Faculty of Biotechnology and Horticulture, University of Agriculture in Krakow, Al. 29 Listopada 54, 31-425 Krakow, Poland; sylwester.smolen@urk.edu.pl (S.S.); dominikgrzanka110@gmail.com (D.G.); 2Department of Plant Product Technology and Nutrition Hygiene, Faculty of Food Technology, University of Agriculture in Krakow, Balicka 122, 30-149 Krakow, Poland; lukasz.skoczylas@urk.edu.pl

**Keywords:** iodine, vanadium, biofortification, 5-iodosalicylic acid, 3,5-diiodosalicylic, 2-iodobenzoic acid, 2,3,5-triiodobenzoic acid, foliar nutrition

## Abstract

A human’s diet should be diverse and rich in vitamins, macro- and microelements essential for the proper functioning of the human body. Globally, a high percentage of the human population suffers from malnutrition, deficiencies of nutrients and vitamins also known as the problem of hidden hunger. This problem it is not only common in poor countries, but also occurs in developed countries. Iodine is a nutrient crucial for the proper functioning of the human and animal body. For plants, it is referred to as a beneficial element or even a microelement. The design of the biofortification experiment was determined on the basis of the interaction of iodine and vanadium (synergistic interaction in marine algae), where vanadium-dependent iodoperoxidase catalyzes apoplastic oxidation of iodine, resulting in high efficiency of iodine uptake and accumulation in brown algae (Laminaria digitate). Three independent experiments (Exp.) were carried out with the foliar application of vanadium (V) and iodine (I) compounds. The main differences between the experiments with the adapted proper corn biofortification method were the different application stage between the individual experiments, the application intervals and the dose of the iodine–vanadium compound. In each experiment, the accumulation of iodine and vanadium in the grain was several times lower than in the leaves. The combination iodine and vanadium significantly increased the accumulation of iodine in the grain in the case of applying V with inorganic iodine compounds, and a decrease in the accumulation of I after applying V with organic iodine compound —especially in Exp. No. 3. In grain, the highest content of I^−^, IO_3_^−^ was in combination with the application of 2-iodobenzoic acid (products of its metabolism). In most of the tested combinations, vanadium stimulated the accumulation/synthesis of exogenous/endogenous 5-iodosalicylic acid (5ISA) and 2-iodobenzoic acid (2IBeA), respectively, and decreased the content of 2,3,5-triiodobenzoic acid (2,3,5-triIBeA) in leaves and grains. The tested compounds I and V and the combinations of their application had a diversified effect on the vitamin C content in the grains. Vanadium in the lower dose of 0.1 µM significantly increased the sugar content in the grain.

## 1. Introduction

Iodine is an essential element necessary for the proper development and functioning of the human and animal organism. Its major role is related to the function it plays in the proper functioning of the thyroid gland [1]. Iodine is a substrate necessary for the synthesis of thyroid hormones, thyroxine-T4 and triiodothyronine-T3 [1,2,3]. Pregnant women are particularly at risk of deficiency of this element, and at the same time the most needy of it. Iodine deficiency during pregnancy is associated with health consequences for the baby, such as impaired fetal brain development, leading to wider and irreversible changes such as cretinism disease. In some cases, insufficient daily dose of iodine in pregnant women can lead to infertility or miscarriage [4,5]. Another health consequence of iodine deficiency is the so-called hypertrophy of the thyroid gland, endemic goiter and increased probability of developing thyroid and stomach cancer [6,7]. The daily dose of iodine is 200–250 µg for pregnant women, 150 µg for adults, 90–120 µg for children from 5–12 years of age, under 5 years old 90 µg, respectively [1,6,7,8].

Around 2 billion people in the world suffer from iodine deficiency in the diet (hidden hunger of this microelement), and about 50 million are diagnosed with an affliction caused by this microelement deficiency [5,9]. The problem with insufficient iodine intake in the diet and the consequences associated with it occur in economically developed countries such as England, Germany, Australia and Italy [10,11]. Even though iodization has shown a great effect, iodine deficiency often persists. Moreover, WHO restrictions on limiting the consumption of table salt (iodization of table salt as one of the main programs to challenge the deficiency of this element in the diet), justified by the possibility of increasing hypertension (especially in pregnant women), initiated many studies and programs on alternative methods of implementation iodine in the human diet; one of them is the biofortification of crops [12,13]. Agrotechnical methods, soil fertilization, foliar application or genetic methods (plant breeding) also constitute cost-effective biofortification strategies [9,14,15,16]. A balanced diet is the basis for supplementing all the macro-/microelements and vitamins needed for proper functioning and the core of this balanced diet are vegetables, fruits and grain products [17,18].

Researches on the methods of effective enrichment of plants with iodine takes into account the type of application (foliar application, soil application, fertigation in field and soilless crops, hydroponics), the type of compound applied—inorganic compounds (such as KI, KIO_3_) and organic compounds (2IBeA, 5ISA, 3,5-di-ISA, 4IBeA)—and plant type in terms of iodine enrichment into edible parts (fruit, grain, leaf, root) [19,20,21,22]. The research objectives and research hypotheses as well as the selection of these factors are different in individual research works. The highest and most effective degree of iodine enrichment is achieved by leafy vegetables [23,24,25]. Corn, wheat and rice can also be effectively biofortified with iodine, the achievement of which was shown by Cakmak et al. 2017 [22].

Information and current knowledge about the largest reservoir of iodine on Earth, which is seawater and the organisms that live in it, was a base for further study on the possibility of implementing that mechanism on terrestrial plants and combining it with sweet corn biofortification. Iodine volatilization from both the marine and terrestrial environments is a major component of the iodine biogeochemical cycle. There are two pathways of iodine volatilization in the marine environment—photochemical and the dominant biological one from micro- and macro-algae [26,27,28]. The capture and accumulation of iodine by algae is mediated by the enzyme vanadium-dependent haloperoxidase (vHPO) [29]. Vanadium-dependent haloperoxidase oxidizes halides and participates in the synthesis of organohalogen [28,30]. Vanadium-dependent haloperoxidase, whose prosthetic group is occupied by vanadium (vHIPO), improves iodine binding by catalyzing the oxidation of I^-^ to more lipophilic compounds (iodine(I) acid) HIO and subsequently molecular I_2_. These molecules easily diffuse across cell membranes into the cytosol. The process of further reduction of HIO or I_2_ to I^−^ in the apoplast is not yet well known, same as the presence of vanadium-dependent iodoperoxidase in higher plants [29,31,32]. Smoleń et al. 2020 conducted research on vanadium-dependent iodine peroxidase activity in lettuce plants [20]. The level of vHPO activity in corn plants at an early stage of their development after soil application of ammonium metavanadate together with organic (5ISA, 2IBeA) and inorganic iodine compounds (KI, KIO_3_) was also tested by Grzanka et al. [33].

Sweet corn grain is rich in nutritional value, it contains large amounts of protein and most vitamins and microelements. The valuable components of sweet corn grains also include such microelements as selenium, chromium, zinc, copper, nickel and iron. It is reasonable to add such an important micronutrient as iodine to this list, since maize for dry grain is one of the three most important cereals worldwide for more than 200 million people [34].

The aim of the research was to obtain an effective level of iodine in corn grain and to evaluate a better method of iodine foliar biofortification with the combined use of inorganic (KI, KIO_3_) and organic iodine compounds (5ISA, 2IBeA) with vanadium. Moreover, the aim was to determine the effectiveness of iodine enrichment of sweet corn grains by foliar fertilization at various stages of plant development and expand knowledge about the content of iodosalicylates and 3,5-diiodosalicylic acid, iodobenzoates, 2,3,5-triiodobenzoic acid and iodide and iodate after foliar application of organic and inorganic iodine compounds to create a basis for iodine synthesis/transformation in maize plants to answer the question of whether a land plant like corn will be able to create an iodine uptake mechanism stimulated by the application of exogenous vanadium.

## 2. Results

### 2.1. Biometrical Parameters and Yield of Corn Cob

In all three experiments, the foliar application of iodine and vanadium compounds in each of the doses used did not have a statistically significant effect on cob yield, average weight of one cob, corn cob length or maximum diameter of the corn cob, as compared to the control (Table 1, Table 2 and Table 3). However, in each of experiments, the addition of vanadium showed the same trend of increasing yield when combined with KIO_3_ with both doses of vanadium. In experiment No. 1 and No. 2, the higher dose of vanadium showed a trend to obtain lower yield in combination with 2IBeA compared to only the 2IBeA application, and a higher yield with the 5ISA + V application compared only with the 5ISA application. According to the analysis, only the use of vanadium in combination No. 1 and No. 2 with a higher dose of vanadium (V2) gave a trend of lower yield compared to the control, in No. 3 the same effect was shown at the lower dose of vanadium (V1). For a separated application of iodine compounds alone in all the performed experiments a similar yielding tendency could not be determined.

### 2.2. Iodine Accumulation in Sweet Corn Plants

In all three experiments, foliar application of organic and inorganic iodine compounds used separately and together with vanadium significantly increased the iodine content in sweetcorn leaves and grains compared to the control (Figure 1A–F).

Analyzing the tested iodine compounds (excluding the combined application of iodine with vanadium), the highest iodine content in sweet corn grains in experiment No. 1 was obtained after the application of the inorganic compound KI (4.8 times higher than in the control), in experiment No. 2 and 3, after 2IBeA application, the contents were 52 and 20 times higher, respectively, than in the control (Figure 1A–C). The level of iodine accumulation in grains after the application of 2IBeA in experiment No. 2 was 1.5 times higher than the level of iodine accumulation after application of KI in experiment No. 1. The iodine concentration in grain after the application of 2IBeA in experiment No. 1 was 1.7 times lower than in experiment No. 2 and 2.9 times lower than in experiment No. 3. In all three experiments, the highest level of iodine accumulation in grains was found in experiment No. 3 after 2IBeA application (Figure 1A–C).

The vegetative parts of sweet corn (leaves) were distinguished by a higher level of iodine accumulation than grains, regardless of the form of iodine applied (Figure 1D–F). The greatest differences between the iodine content in leaves and grains were observed in experiment No. 2 (on average 25 times higher iodine content in leaves than in grain), and the lowest in experiment No. 1 (an average of 6 times higher iodine content in leaves than in grains). Analyzing the tested iodine compounds (excluding combinations with the application of iodine with vanadium), the highest iodine content in sweet corn leaves in experiment No. 1 was obtained after the application of the inorganic compound KI (2.3 times more than in the control), and in experiment No. 2 it was also KI (9.5 times more than in the control). In the experiment No. 3, the highest degree of iodine accumulation in leaves was determined after application of 2IBeA (12.4 times more than in the control) (Figure 1D–F). The lowest degree of accumulation in leaves compared to the other applied iodine compounds in experiments No. 1 and No. 2 was obtained after 5ISA application. In turn, in experiment No. 3, the application of KI (excluding combinations with the application of iodine with vanadium) resulted in the lowest degree of iodine enrichment of the leaves in relation to other iodine compounds, both organic and inorganic.

#### 2.2.1. The Interaction of Iodine with Vanadium in Individual Parts of the Plant

In experiment No. 1, application of vanadium (in doses of 0.1 µM and 1.0 µM V) together with iodine significantly influenced the iodine content in corn grains only when used in conjunction with KIO_3_ (KIO_3_ + V1, KIO_3_ + V2 versus KIO_3_). There was an 8% and 18% increase in iodine accumulation in grains compared to KIO_3_, respectively (Figure 1A). The combination of KI, 5ISA and 2IBeA with vanadium (in both doses excluding 2IBeA + V1) reduced the iodine content in corn grains in experiment No. 1 (Figure 1A). In experiment No. 2, no statistically significant differences were found in the accumulation of iodine in grains in combinations with and without the addition of vanadium after the application of KI, KIO_3_ and 5ISA. In this experiment, the higher dose of vanadium (1.0 µM V) in combination with 2IBeA resulted in a 7% reduction of iodine content compared to the combination with 2IBeA application without vanadium (Figure 1B). Using of vanadium at a dose of 0.1 µM V in experiment No. 3 in combination with inorganic iodine compounds increased the accumulation of iodine by 9% for KI + V1 in relation to KI and by 19% for KIO_3_ + V1 in relation to KIO_3_. In the case of combined application of vanadium with organic vanadium compounds, a decrease in iodine content in the grain was observed, by 21% for 5ISA + V1 versus 5ISA, and by 9% for 2IBeA + V1 versus 2IBeA, respectively (Figure 1C).

In the leaves in experiment No.1, the combined foliar application of iodine and vanadium in the form of ammonium metavanadate had no statistically significant effect on the iodine accumulation for the combination with 5ISA versus 5ISA + V1, 5ISA + V2. The lower dose of 0.1 µM V vanadium used with 2IBeA also had no statistically significant effect on the accumulation of iodine in the leaves. On the other hand, after using a higher dose of vanadium with 2IBeA (2IBeA + V2), a 20% reduction of iodine accumulation in leaves was observed compared to 2IBeA applied alone. The stimulating effect of a lower dose of vanadium was found in the combination KIO_3_ + V1 vs. KIO_3_ (increase by 11%). For KIO_3_ + V2 there was a 4.5% decrease in the iodine content in the leaves in relation to KIO_3_. In the objects KI + V1 and KI + V2, the iodine content in the leaves was lower by 15% and 18% compared to the application of KI alone (Figure 1D).

In experiment No. 2 no beneficial effect of vanadium on iodine accumulation after the use of KI + V1 and KI + V2 compared to the only application of KI (decrease in iodine content in leaves by 29% and 44%, respectively) (Figure 1E) was shown. In the other tested compounds, the lower dose of metavanadate had a stimulating effect by increasing the iodine content in the leaves: by 22% for KIO_3_ + V1 vs. KIO_3_, by 37% for 5ISA + V1 vs. 5ISA as well as by 60% 2IBeA + V1 vs. 2IBeA (Figure 1E).

In experiment No. 3, where only one lower dose 0.1 μM V of ammonium metavanadate was used, a statistically significant increase in the iodine content in the leaves was found after the combined application of KI + V1 vs. KI by 13%, by 40% after the use of KIO_3_ + V1 vs. KIO_3_ and by 7% as a result of foliar application of 2IBeA + V1 vs. 2IBeA. Only the combined application of 5ISA + V1 caused a 14% decrease in the accumulation of iodine in the leaves versus 5ISA (Figure 1F). The level of iodine accumulation in the leaves in experiments No. 2 and No. 3 were several thousand micrograms higher than in experiment No. 1.

#### 2.2.2. The Content of Iodide, Iodates Ion and Organo-Iodine Compounds in Sweet Corn Grain and Leaves

All tested iodine compounds increased the content of iodide (I^−^) ions in corn grains in experiment No. 3. The highest, 4.7-fold increase in comparison to the control, was observed after the application of 2IBeA without vanadium. The content of iodide (I^−^) ions decreased 1.5 times in combination with the addition of vanadium 2IBeA + V1 compared to the solo application of 2IBeA (Figure 2A). The addition of vanadium in the case of KI application caused a statistically significant increase in the content of I^−^ ions compared to the application of KI without vanadium. In turn, the level of IO_3_^−^ accumulation in corn grains was the highest in the application of KIO_3_ and 2IBeA without vanadium. The addition of vanadium to these compounds resulted in a 2.4-fold reduction in the content of IO_3_^−^ for combination KIO_3_ + V1 vs. KIO_3_ and 2.5-fold for 2IBeA + V1 vs. 2IBeA. In the combination of 5ISA + V1 a statistically significant increase (1.4-fold) in the level of IO_3_^−^ content in the grain was shown compared to the combination without vanadium (5ISA) (Figure 2B).

The highest 5ISA content in grain and leaves was found after the foliar application of 5ISA + V1 compared to the control and other combinations (Table 4). It should be noted that the content of 5ISA in grain and leaves after the application of this compound without vanadium was lower than in the combination of 5ISA + V1. The lowest 5ISA content in grain was obtained in the control and the facility with the 2IBeA application. The highest 3,5-diISA content in sweet corn grain was recorded in the grain of the control plants. It was 3.6 times higher than in the combination with the lowest 3,5-diISA content, i.e., after the application of KIO_3_. On the other hand, the tendency of the 3,5-diISA content in the leaves was different—the leaves of the control plants contained the least 3,5-diISA, and the most leaves after the application of KIO_3_ + V1. In the tested combinations KIO_3_ vs. KIO_3_ + V1, KI vs. KI + V1 and 5ISA vs. 5ISA + V1, an increasing 3,5-diISA content was noted, which was a significant effect of vanadium addition. The 2IBeA content in leaves and grains was the highest after foliar application of this compound. However, a synergistic effect of vanadium on the accumulation of 2IBeA in grain and an antagonistic effect of vanadium on the accumulation of 2IBeA in the leaves was noted. The higher accumulation of 2IBeA in the grain was found in the case of the combined application of 2IBeA + V1 versus 2IBeA. In the case of the KI and 5ISA application, their combined use with vanadium compared to the application without vanadium resulted in a reduction in the 2IBeA content in the grain, and for 5ISA + V1 versus 5ISA also in the leaves. The content of 2,3,5-triIBeA in corn grain and leaves was several hundred times lower than 5ISA, 3,5-diISA and 2IBeA. The highest content of 2,3,5-triIBeA in grain was recorded after application of 2IBeA, and in leaves after application of V1. In grain and leaves simultaneous foliar application of KI + V1 versus KI; and KIO_3_ + V1 versus KIO_3_ and 5ISA + V1 versus 5ISA but only in leaves; as well as 2IBeA + V1 versus 2IBeA only in grain, it caused a decrease in 2,3,5-triIBeA content.

### 2.3. Vanadium Content in Sweet Corn Plants

The vanadium fertilization applied separately as well with all iodine compounds showed a significant effect on the vanadium content on the leaves and grain of sweet corn (Figure 3A–F). In experiments No. 1 and No. 2, with the increase in the dose of used vanadium, the content of this element in sweet corn leaves and corn grains increased (Figure 3A,B,D,E). However, only when vanadium was applied in combination with 5ISA were there no statistically significant differences in terms of V content in the grain between the combinations 5ISA + V1 and 5ISA + V2 in both experiment No. 1 and No. 2 (Figure 3A,B). The vanadium content in the leaves was on average about 30 times higher than in the seeds in experiment No. 1. and about 100 times higher than in grains in experiment No. 2. In experiment No.1, the highest content of vanadium in the grain was recorded in the combination KIO_3_ + V2, while in experiment No.2, this occurred in the combination of 2IBeA + V2. The application of vanadium in a higher dose in experiment No. 1 resulted in significantly lower accumulation of vanadium in leaves than the combined application of vanadium with iodine in the combinations KI + V2 vs. V2 by 48% and 5ISA + V2 vs. V2 by 31% (Figure 3D). In experiment No. 2, the leaves of plants from all combinations of iodine + V2 were characterized by a higher content of vanadium than after the application of only vanadium in a higher dose (V2; Figure 3E). In experiment No. 2, for individual combinations, the application of iodine with vanadium at a higher dose (iodine + V2) versus V2 alone, an increased accumulation of vanadium in the leaves was noted by: 50% for KI + V2 vs. V2; 43% for KIO_3_ + V2 vs. V2; 35% for 2IBeA + V2 vs. V2 as well as 21% for 5ISA + V2 vs. V2 (Figure 3E). 

In the experiment No. 3, only the lower dose of ammonium metavanadate V1 = 0.1 µM was applied. In this experiment, there was also a tendency to increase the vanadium content in sweetcorn leaves and grains after the application of vanadium with iodine compounds versus the application of these iodine compounds without vanadium (Figure 3C,F). In the experiment No. 3, the content of vanadium in leaves was on average 20 times higher than in grains (Figure 3C,F). The highest content of vanadium in grains was obtained after application of 5ISA + V1, and in leaves after application of only vanadium without iodine. 

### 2.4. Total Sugars and Vitamin C Content in Sweet Corn Grains

In all carried out experiments (three separate experiments), the content of total sugars (glucose + fructose + sucrose) and vitamin C (L-ascorbic acid) in sweet corn grains significantly depended on applied iodine compound and dose of vanadium (Figure 4A–F see also Appendix A). In experiment No. 1, the highest total sugar content was obtained in the control (Figure 4A), and in experiments No. 2 and 3, after the application of a lower dose of vanadium V1 (Figure 4B,C). In experiment No. 1 and No. 2, all applied iodine compounds had a significantly lower total sugar content compared to the control treatment. The most significant decrease of total sugars content was noted after 2IBeA application (Figure 4A,B). In experiment No. 3, we did not note the same impact of 2IBeA application on total sugar content. Using 2IBeA did not show statistical differences compared to the control (Figure 4C). In experiment No. 2 and 3, the application of 5ISA without vanadium resulted in a significant increase in the content of sugars in the grains.

In experiment No. 1, the highest content of vitamin C in grains was found after foliar application of KIO_3_ + V2 and the lowest after foliar application of V1, V2 and KI + V2 (Figure 4D). In experiment No. 2, the content of vitamin C was the highest after 2IBeA + V2 foliar application and the lowest after 2IBeA + V1 application (Figure 4E). In this experiment, along with the increasing dose of vanadium applied with iodine, the value of ascorbic acid in grains increased after the application of KI, KIO_3_ and 5ISA. The aforementioned relations for the increasing content of vitamin C in grains are as follows: KI + V1 vs. KI increase by 11% for KI + V2 vs. KI increase by 21%; KIO_3_ + V1 vs. KIO_3_ increase by 6%, KIO_3_ + V2 vs. KIO_3_ increase by 25%; 5ISA + V1 vs. 5ISA increased by 12.9% and 5ISA + V2 vs. 5ISA increased by 13%. In experiment No. 3, the application of 2IBeA + V1 to 2IBeA had no effect on the vitamin C content in the grains. In the case of the other applied iodine compounds, the addition of ammonium metavanadate resulted in a statistically significant decrease in the ascorbic acid content in the grains (Figure 4F). In this experiment, the highest vitamin C content in grains was found after the application of KIO_3_ and then KI.

### 2.5. Percentage of Recommended Daily Allowance (RDA) for Iodine (RDA-I) and Hazard Quotient (HQ) in Sweet Corn Grain

Based on chemical analyzes of sweetcorn grain in the milk stage, the percent RDA for I (RDA-I) was calculated in 100 g F.MW. sweet corn grains and as well the hazard ratio (HQ) (Table 5).

Iodine consumption was calculated on the basis of adult consumption of 100 g of sweet corn grains (average body weight 70 kg). Consumption of 100g of sweet corn grains from plants enriched with I and I + V would significantly increase the RDA-I in relation to the control in all three experiments. The consumption of iodine-biofortified corn grains (applied separately and together with V) would allow the human body to be provided with between 2.89% (KIO_3_ application) to 3.79% (KI application) of RDA-I in experiment No. 1, from 0.66% (application KIO_3_ + V2) to 4.62% (application 2IBeA) of RDA-I in experiment No. 2 and from 0.47% (KIO_3_ application) to 6.84% (2IBeA + V1 application) of RDA-I in experiment No. 3 (Table 5). The harmfulness of iodine to the human body would occur if the HQ-I index was ≥ 1.0. In all three experiments, the calculated HQ-I coefficients were significantly lower than 1.0. The highest value of HQ-I occurred after the use of KI (0.00212 HQ-I) in experiment No. 1, 2IBeA (0.00325 HQ-I) in experiment No. 2 and 2IBeA + V1 (0.00527 HQ-I) in experiment No.3.

## 3. Discussion

Corn is a very important element in the diet of people all over the world, regardless of the level of economic development of the countries, and it is an important energy crop [35]. Corn and food products derived from it are of great importance for people intolerant to gluten and with celiac disease. Around 1% of the world’s population suffers from this type of disorder [36,37]. The implementation of iodine enrichment of crops (corn, cereals, vegetables) in agricultural practice would reduce problems with iodine deficiency occurring all over the world [5,7,10]. The last two decades have provided a lot of research on agrotechnical methods of enriching crops with iodine [6,11]. Many of them proved the effectiveness of the iodine compounds used, which were mainly KI and KIO_3_ [22,38]. There had been available results of studies on the biofortification of crops into iodine with the use of organic compounds of this element, including 5-iodosalicylic acid, 3,5-diiodosalicylic acid, 2-iodobenzoic acid, 4-iodobenzoic acid or iodoacetic acid in tomato cultivation [19], lettuce [39,40] and spinach [41]. It is also known that applying iodine in the fertilization of plants in combination with organic stabilizers. In the research of Rangel et al. 2020, the effect of enriching lettuce with iodine was obtained after using the chitosan-I complex (Cs-KIO_3_, Ch-KI) [42].

The studies determined the interaction of two trace elements, iodine and vanadium, in corn plants. This interaction—the participation of vanadium in iodine uptake—is described for several species of sea algae [29,31,43]. All of the tested combinations of iodine and iodine with vanadium in the three carried out experiments significantly influenced the degree of iodine accumulation in leaves and grains of sweet corn. Effective enrichment of sweet corn grains with iodine was demonstrated after foliar application of both organic (5ISA, 2IBeA) and inorganic (KI, KIO_3_) iodine compounds. The highest effect of enriching sweet corn grain with iodine was obtained after the application of the organic iodine compound 2IBeA in two experiments, No. 2 and No. 3 (analyzing only the effectiveness of iodine compounds applied individually, without vanadium). On the other hand, in experiment No. 1, the inorganic KI compound turned out to be the most effective, and the second in line was the organic 2IBeA. The conducted experiments allowed three aspects determining the effectiveness of enrichment of grains with iodine to be documented, without leading to any phytotoxic symptoms of iodine and vanadium on plants. These include the issue of increasing the dose of iodine from 10 µM I (experiment No. 1) to 100 µM I (experiment No. 2 and 3), compressing the application period between subsequent foliar applications treatments, and carrying out foliar application I and V in various phases of plant development. This is the performance of foliar application I and V at the beginning of flowering (experiment No. 3) doubled the level of iodine accumulation in grains compared to the application of these elements in the earlier stage of plant development (experiment No. 2). The grains of sweet corn treated only with 2IBeA in experiment No. 3 contained about 2.6 times more iodine than in the combination of KI from experiment No. 1 and 1.8 times more than after the application of 2IBeA in experiment No. 2. Satisfactory effects of enriching tomato fruits (generative organs) with iodine after 2IBeA application were obtained by Halka et al. 2019 [44]. In the experiment with tomato in the seedling phase after soil application of iodine compounds, both the roots, leaves and shoots after applying KI had the highest content of iodine, followed by plants treated with 2IBeA and 4-IBeA (4-iodobenzoic acid) [19].

From the point of view of an effective implementation program of foliar iodine biofortification, the 100 µM iodine dose used in experiment No. 3 was safe for plants and did not cause phytotoxic symptoms on sweet corn plants. Phytotoxic symptoms were demonstrated in studies with the application of inorganic forms of iodine in rice plants in doses of 10 and 100 µM KI and 100 µM KIO_3_ [45] or while fertigation 2.34 mM KIO_3_ and 3.01 mM KI of corn, barley, potato and tomato [46]. In the experiment with the use of organo-iodine compounds 3,5-diISA (3,5-diiodosalicic acid) and 4-IBeA at a dose of 25 μM I in tomato plants due to fertigation in the seedling phase, after the applied organo-iodine compounds a negative effect on the growth and development of above-ground young tomato plants parts was observed [19]. 5ISA in a dose of 40 μM I led to a significant decrease in the biomass of roots and leaves of lettuce grown in the NFT hydroponic system [39].

The used compounds in all experiments with both a lower dose (No. 1) and a higher dose of iodine (No. 2 and No. 3) did not have a statistically significant effect on the sweet corn yield; similar results after soil and foliar application of KI and KIO_3_ were obtained Cakmak et al. [22]. In the pot tests on sweet corn, no toxic effect was found and no statistically significant effect on the yield and development of young corn plants at a lower iodine dose of 10 µM I and analogously doses of vanadium 0.1 µM and 1 µM V [43]. The iodine content in the leaves in all experiments was on average 14 times higher than in the grains. The higher dose of iodine compounds used significantly increased the content of iodine in the leaves by about 6 times in experiment No. 2 and No. 3 compared to No. 1. Vegetative parts of plants (leaves, shoots) accumulate higher amounts of this element than the generative parts. Similar results were obtained in studies on rice [22,45], sweet corn [43], green beans [47] and plum trees [21].

The transport of iodine from the leaves to the corn grains after foliar application would have to be via the phloem. The phloem transport of iodine has been confirmed in many studies on the effectiveness of the accumulation of this element in the generative parts [21,38,48]. Nevertheless, xylem transport of iodine is much more efficient. Hurtevent et al. (2013) pointed to the fact of the relative mobility of iodine in the phloem which was proven in the research by Zou et al. 2019 [49,50]. Foliar iodine application (inorganic KIO_3_) in the wheat grain filling stage created an available pool of iodine in the leaf tissues, causing phloem transport to the grain. In this phase, there is an intense transfer of photo assimilates to the seeds, and the activation of the phloem transport [50]. The presented research in experiment No. 3 also confirmed the described effects of effective phloem transport as a consequence of more effective accumulation of iodine in grains.

The combined application of iodine with other micronutrients such as Zn, Fe, Se has been described in studies on wheat and rice [9,50]. Foliar application of I, Zn, Se and Fe combined (in most locations where it was performed) resulted in less iodine accumulation than the same dose of iodine applied individually [50]. In the case of vanadium, the interaction of vanadium with iodine, which could be the stimulation or antagonism of this element, is not known in higher plants yet. In a group, brown algae, enzymes as vanadium-dependent haloperoxidases (vHPO) play a key role in the uptake and accumulation of iodine (as well halides metabolism of Cl and Br) [29,31]. They are mainly responsible for the capture of iodine from the water, the synthesis of hydrogen halides in marine algae and antioxidant defense [32,51]. The studies on the interaction of iodine and vanadium in higher plants was conducted, among others, on lettuce by Smoleń et al. [20,40] and in sweet corn by Grzanka at al. [33,43].

In all experiments, in sweet corn leaves the stimulating effect of vanadium on iodine accumulation was preeminent. In experiments No. 1, No. 2 and No. 3, the application of KIO_3_ + V1 was statistically significantly more effective in terms of iodine accumulation than the application individually of KIO_3_. For 2IBeA in experiment No. 1 and No. 3, iodine accumulation in 2IBeA + V1 was at a similar level of significance with 2IBeA; in experiment No. 2 both doses (0.1 µM and 1 µM) of vanadium significantly increased I accumulation compared to the individually application of 2IBeA. In experiment No. 1, combined application 5ISA with vanadium did not show any impact on the accumulation of iodine. In experiment No. 2, after using the combination of 5ISA + V1, the iodine content in the leaves was significantly higher than after application of only 5ISA. In turn, in experiment No. 3, only the combination of 5-iodosalicylic acid with vanadium resulted in a decrease in the accumulation of iodine in the leaves (and in grain) compared to the individually applied of 5ISA.

A similar effect was obtained in a pot experiment with soil application of 5ISA with two doses of vanadium (0.1 µM and 1 µM) in sweet corn cultivation [43]. The combined application of iodine and vanadium in a hydroponic system with organic and inorganic iodine compounds gave a variable effect. Lettuce grown in the hydroponic system in the combination of 5ISA + V and for 3,5-diISA + V resulted in a significantly higher level of iodine accumulation in the roots than the individual application of organic compounds 5ISA and 3,5-diISA. In the lettuce leaves the stimulating effect for both compounds used in hydroponics and as well in peat substrate was not observed. On the other hand, in the mineral soil, lettuce leaves had a statistically significantly higher iodine content in/after treatment with 3,5-diISA + V vs. 3,5-diISA [40].

The combined application of 2IBeA with vanadium resulted in a statistically significant decrease in the accumulation of iodides and iodates compared to the individual application of 2IBeA. On the other hand, the combination of KI + V1 increased the accumulation of iodides in the grain of sweet corn compared to the solo application of KI. The combined application of KIO_3_, 2IBeA, 5ISA with vanadium resulted in a significant increase in the synthesis/accumulation of 5ISA and 3,5-diISA iodosalicylates in grain. In the leaves, this relationship is consistent with the combined application of the vanadium from KI, KIO_3_ and 5ISA. The combined application of vanadium with all iodine compounds, both organic and inorganic, significantly reduced the accumulation of 2,3,5-triIBeA in leaves and grains of sweet corn. The literature indicated that this organo-iodine compound is an auxin inhibitor [52]. The results of the research based on a one-year experiment with a lower dose of vanadium (No. 3 year 2020) confirm the stimulating effect of vanadium (in a dose of 0.1 µM) on the accumulation of iodosalicylates (5ISA, 3,5-diISA) in the leaves and grains of sweet corn in combination with organic and inorganic iodine compounds. Research on tomato [44] and lettuce [20] also confirm the occurrence of iodosalicylates and iodobenzoates in control plants. After foliar application of iodosalicylates and iodobenzoates, their accumulation in sweet corn leaves and grain increased. Similar results were found in studies on tomato and lettuce [20,44].

Transport of vanadium from the roots to the higher parts of plants, mainly to the generative organs, is limited. The highest degree of accumulation of this element is in the roots [53,54]. It is related to the process of vanadium biotransformation during the uptake of vanadium by the roots. This biotransformation consists in the reduction of pentavalent vanadium, which easily oxidize ketones, aldehydes, catechols, sulfhydryls and olefins placed in the cell wall even at pH 7. As a result, vanadium is retained by root tissues and vanadium (V) is reduced to the quadrivalent form of vanadium (IV) [55].

In the conducted experiments No. 1 and No. 2 (with two applied doses of vanadium 0.1 and 1 µM), an increase in the applied dose of vanadium increased the accumulation of this element in leaves and corn grains. The effect of increased accumulation of vanadium with the increasing dose of this element was confirmed in the cultivation of pepper [56], chickpea [57] and beans [54]. In experiments No. 1 and No. 2, the application of KI and 5ISA stimulated the accumulation of vanadium by increasing its content in the leaves compared to the application of only V2 (vanadium in a higher dose) without iodine. At the same time, V2 significantly reduced iodine accumulation in combination with 5ISA and KI. A similar effect was obtained in studies on sweet corn with the application of iodine and vanadium compounds to the soil [43]. In the early development stages of sweet corn, the accumulation of iodine in the roots was highest in combination with the application of solo ammonium metavanadate at a dose of 0.1 µM [33]. Vanadium is determined as a beneficial element to higher plants. The stimulating effect for plants ranges from 1–10 µg L^−1^; higher doses have a phytotoxic effect on plants [55,58,59]. Vanadium furtherance the nitrogen fixation process in soils with deficient in molybdenum, low doses stimulate the synthesis of chlorophyll [59,60]. In the carried-out experiments, vanadium doses of 0.1 µM and 1 µM did not cause any phytotoxic effect on plants. Vanadium did not affect the sweet corn yield in a statistically significant way.

The content of total sugars and vitamin C was mainly formed by the weather conditions in the year of the experiment, and it was characterized by a huge diverseness between the studied combinations. In experiment No. 1 and No. 2, the grains had the lowest total sugar content after the foliar application of 2IBeA. The highest total sugars content was in the control for experiment No. 1, and for No. 2 and No. 3 with the application V1 (0.1 µM). In the experiments No. 1 and No. 2, the addition of vanadium to KIO_3_ increased the content of vitamin C in the grains. In No. 3, the exclusive application of KIO_3_ significantly increased the content of ascorbic acid. In the experience of lettuce the total sugar content and vitamin C was mainly determined by type of applied iodine compounds (organic or inorganic iodine), as well largely the dose determined the variability in sugar and vitamin C content. After the application of 5ISA in a dose of 8.0 μM was obtained the lowest level of vitamin C. The highest total sugar content was recorded in plants grown in a nutrient solution containing KIO_3_ + SA. 5ISA in the dose of 40 μM resulted in a higher total sugar content in the lettuce compared to the lower doses (1.6 μM and 8 μM) [39]. The application of KI and KIO_3_ increased the content of vitamin C in Opuntia ficus-indica var. Copena V1 almost two times [61] as well in water spinach [41].

An important issue for the implementation of biofortified plants is regular monitoring of iodine intake status to detect excessive intakes. Some data emphasize that healthy adults who are iodine sufficient are curiously tolerant to iodine intake even 1000 µg·day^−1^. Appropriate iodine intake is the most important for people in chronic iodine deficiency because swift increase of iodine may cause thyroiditis or hyperthyroidism [10]. Upper levels of biofortification of the edible parts plants has to be an I of HQ < 1.0, which indicates a safe level for the consumer. In the presented study, this indicator HQ was not exceeded value 1.0 in all the conducted experiments. In the third experiment after the application of organic iodine compound 2IBeA and 2IBeA + V1, the I-RDA is the highest, so if the average daily intake per adult is 200 g, we can provide approximately 14% of the recommended daily iodine intake.

## 4. Materials and Methods

### 4.1. Plants Material, Treadment and Meteorological Data

In 2018–2020, three independent experiments were carried out with the field cultivation of sweet corn (*Zea mays* L. subsp. Mays Saccharata) cv. “Złota Karłowa” on the horticultural farm in southern Poland (50° 16′53.7″ N 19° 47′43.5″ E). The effectiveness of iodine biofortification of corn grains after foliar application of inorganic and organic iodine compounds applied separately and in combination with vanadium in the form of ammonium meatavanadate was investigated. In experiment No. 1 and No. 2 (Table 6), two doses of vanadium was tested, 0.1 µM V and 1.0 µM V, in combination with iodine and was applied at the same plant growth phases 32–61 BBCH. The difference between experiment No. 1 and No. 2 consisted of the fact that in experiment No. 1 (in 2018), iodine was applied at a dose of 10 µM I, and in experiment No. 2 (in 2019), it was applied at a dose of 100 µM I. In experiment No. 3 (in 2019 and 2020), only one lower dose of vanadium 0.1 µM V was used as foliar treatment in combination with the tested iodine compounds at a dose of 100 µM I (Table 7). In experiments No. 1 and No. 2, the application of iodine and vanadium compounds started from the stage of stem development (shoot elongation), i.e., in the 2 nodes phase—BBCH 32. The period between successive applications was on average 14 days and ended in the 61 BBCH phase (i.e., 14 days before harvesting the corn cobs). In experiment No. 3, the application of iodine and vanadium compounds in the form of a foliar spray was started in the phase of visible stamens in the spikelets of the middle part, when the cob emerged from the leaf sheath, i.e., from the BBCH 61 phase. The period between successive applications was 3 days. In all three experiments, four treatments of foliar application of iodine and vanadium compounds were performed. In each experiment, the following iodine compounds were applied: KI, KIO_3_, 5-iodosalicylic acid (5ISA) and 2-iodobenzoic acid (2IBA). The detailed scheme of the three experiments and the tested combinations are presented in Table 1 and Table 2.

In each experiment, there were 60 plants for one combination, i.e., 4 replications, 15 plants for one replication—the experiments were carried out in a randomized system, separately for each experiment. Treatments were randomized over the entire experiment area of each experiment. The distance between the rows was 67.5 cm, and the distance between the plants in the row was 15 cm. In each year of the study, no fodder corn was cultivated within a few kilometers from the experimental field. The same pre-sowing fertilization was used in all experiments: 130 kg N⋅ha^−1^, 80 kg P_2_O_5_⋅ha^−1^, 170 kg K_2_O·ha^−1^, 40 kg MgO·ha^−1^. Chemical protection of plants against weeds, diseases and pests was also followed in accordance with the plant protection program in force in Poland. In all experiments, corn cobs were harvested at full milk maturity of the granuloma (BBCH 75). After harvesting, the casings were removed from the corn cobs, sequentially counted and weighed to evaluate the yield size. All the corn cobs from each plant/each replicate were collected for chemical analysis, and then manually obtained grains (half of the grains from each corn cob). Simultaneously with the collection of the corn cob, one leaf from five plants from each replicate was collected for chemical analysis—the leaves located directly under the corn cob were collected.

The average daily temperature in the period from April to September in 2018 differed from the average daily temperature in 2019 and 2020 (in which it was similar) (Table 8). In 2018, the hottest month was July, while in 2019 the hottest month was June, and in 2020 the hottest month was August. April was the warmest in 2018, and in 2020 the average monthly temperature was halved. The largest difference in the average daily temperature between 2018 and 2019 was in May, and between 2019 and 2020 in June. The sum of rainfall in 2019 in the period from April to September was the highest compared to 2018 and 2020. Compared to 2020, it was higher by 64.8 mm, and compared to 2018 by 93.3 mm. The sum of rainfall in 2018 and 2020 was evenly distributed in individual months from April to September. In turn, in 2019, May and August were the months with the highest rainfall, and June and July with the lowest rainfall, respectively 14.5 mm and 22.0 mm.

### 4.2. Analysis of Fresh Samples of Leaves and Grains

The samples of fresh grain (milk stage) were then homogenized, and total sugars, as a sum of glucose, fructose, and sucrose, were extracted with boiling 96% ethanol (Destylernia ‘Polmos’ Sp. z o.o., Kraków, Poland). The content of fructose, glucose and sucrose and their sum as total sugars was assessed by using the capillary electrophoresis technique with the PA 800 Plus system (Beckman Coulter, United States). Capillaries of ø 50 mm and total length of 60 cm (10 cm for detection) were used. A positive power supply of 15 kV was applied, and the temperature was set at 25 °C. The running buffer solution comprised 20.0 mmol/L sorbic acid, 0.20 mmol/L CTAB, and 40 mmol/L NaOH, pH 12.2. [62].

The content of L-ascorbic acid in fresh grains was analyzed by capillary electrophoresis after the homogenisation of 20 g samples in 80 cm^3^ of 2% oxalate acid (puriss. p.a., Avantor Performance Materials) and further centrifugation for 15 min at 4500 rpm, 5 °C. The supernatants were filtered through a 0.25 µm cellulose acetate membrane filter and analyzed using a PA 800 Plus capillary electrophoresis system (Beckman Coulter, Indianapolis, IN, USA) with diode array detector (DAD) detection. Capillaries of 50 µm i.d. and 365 µm o.d. and those of a total length of 50 cm (40 cm to detector) were used. A negative power supply of 25 kV was applied. The running buffer solution was prepared as proposed by [63], containing 30 mM NaH_2_PO_4_ (puriss. p.a., Avantor Performance Materials), 15 mM Na_2_B_4_O_7_ (puriss. p.a., Sigma-Aldrich, Darmstadt, Germany) and 0.2 mM cetyltrimethylammonium bromide (CTAB) (puriss. p.a., Sigma-Aldrich) (pH 8.80).

### 4.3. Analysis of Dry Samples of Leaves and Grains

Fresh samples of leaves and grain were dried at 70 °C (48 h) in a laboratory dryer with forced air circulation. Dried samples of leaves and grain were ground in a laboratory mill and stored in a plastic bag until the analyses of iodine and vanadium contents were carried out. The dry weight content in these samples was determined using the oven-drying method at 105 °C.

To determine iodine content, the PN-EN 15111-2008 method was used with the modifications described by [64] using ICP-MS/MS (iCAP TQ ICP-MS ThermoFisher Scientific, Bremen, Germany). The concentrations of V were determined using the ICP-OES spectrophotometer (Prodigy Spectrometer, Leeman Labs, New Hampshire, MA, USA) after microwave digestion in 65% super pure HNO_3_. Plant samples of 0.5 g of dry material were placed in 55 mL TFM modified polytetrafluoroethylene (PTFE) vessels and digested in 10 mL of 65% HNO_3_ using a CEM MARS-5 Xpress (CEM World Headquarters, Matthews, NC, USA) microwave digestion system [65].

Speciation of I, i.e., iodides and iodates, was analyzed in the dried samples of grains (only from experient No. 3 in 2020) using high-performance liquid chromatography (HPLC)–ICP-MS/MS. The content of these two I ions was measured using a modified extraction procedure described by [66], whereby 0.05 g of air-dried, ground plant samples were mixed with an extraction solution containing 4 cm^3^ of 25% TMAH (Sigma-Aldrich Co., LLC) and 10 cm^3^ 0.1 M NaOH (Chempur, Piekary Śląskie, Poland), in 1 dm^3^ of demineralized water. The samples were placed in 7-mL polypropylene tubes, whereupon 5 mL of the extraction mixture was added. Once mixed, the samples were incubated for 1 h at 50 °C in an ultrasonic bath and then cooled to approximately 20 °C, mixed thoroughly, and centrifuged for 15 min at 4500 revolutions/min. The supernatants were filtered through a 0.22 μm syringe filter. The content of I ions in filtered samples was analyzed using HPLC—ICP-MS/MS. For I^–^ and IO_3_^–^ speciation forms, HPLC (Thermo Scientific Ultimate 3000; Thermo Fisher Scientific, Bremen, Germany) was coupled to ICP-MS/MS (iCAP TQ). This method employed a strong anion exchange column (Thermo Scientific; Dionex IonPac AS11 [4 × 250 mm]) and a pre-column (Thermo Scientific; Dionex IonPac AG11 [4 × 50 mm]). The column temperature was set to 30 °C. Demineralized water, 50 mM NaOH, and 0.5% TMAH were used as eluents. To separate both I ions, a mobile phase was used, containing 2.5 mM NaOH and 0.125% TMAH with an isocratic flow. The flow rate was 1.5 mL/min, with an injection volume of 10 μL, and total analysis time of 7 min. The 127I.16O isotope of I was determined, using the S-TQ-O2 mode. Standards were prepared through the dissolution of KI and KIO_3_ (Sigma-Aldrich Co., LLC) in demineralized water.

Grains only from experient No. 3 in 2020 were analyzed for iodosalicylates and iodobenzoates [5-iodosalicylic acid (5ISA) and 3,5-diiodosalicylic acid (3,5-diISA); 2-iodobenzoic acid (2IBeA), 2,3,5-triiodobenzoic acid (2,3,5-triIBeA)], using the liquid chromatography LC–MS/MS system after extraction with 75% ethanol [20]. Measurements were made using the HPLC Ultimate 3000 system (Thermo Scientific) and an LC-MS/MS: 4500 Qtrap, Sciex spectrometer. Chromatographic separation was carried out on a Luna 3 μm phenyl-hexyl 100 Å (150 × 3 mm, internal diameter 3 μm) column (Phenomenex, Torrance, CA, United States). Electrospray ionization in negative ion mode was used. MS/MS was performed for quantitative analysis. The LC-MS/MS system was controlled using Analyst 1.7 with HotFix 3 software, which was also used for data processing [20].

The percentage of the RDA for I (RDA-I) and Se (RDA-Se) in 100-g portions of the sweet corn grain was calculated based on chemical analyses of the corn grain as well as the hazard quotient (HQ). The intake of I and Se was calculated based on the consumption of 100 g of fresh sweet corn grain by adults (average of 70 kg body weight). All of the calculations were based on the methods described by Smoleń (2019) [67].

### 4.4. Statistical Analysis

The statistical analysis was performed by using Statistica 13.1 PL programme. All of the data of plant analysis were examined using analysis of variance (ANOVA). Statistically significant differences were assessed by the post-hoc Duncan’s test. *p* values less than 0.05 were considered as statistically significant.

## 5. Conclusions

Compared to the control, there was a tendency to increase the yield of maize cobs after the application of KIO_3_ with vanadium. The most effective level of enrichment of sweet corn grain was achieved in experiment No. 3. The three experiments carried out allowed for the determination of the safest, and at the same time, most effective dose of iodine (100 μM) for sweet corn grain, optimal application time corn development phase BBCH 61–69 and application interval. The highest accumulation of iodine in the grain after foliar application of 2IBeA and the clearly visible positive effect of vanadium on the accumulation of iodine in the leaves with inorganic compounds KI and KIO_3_ were observed in experiment No. 3. Experiments confirmed the low mobility of vanadium in plants; the content was the highest in the vegetative plant parts compared to the generative (grain). An interesting aspect was that after the application of the organic iodine compound, 2IBeA was the largest accumulation of I^−^ in the grain, and after the application of this foliar organic compound, the synthesis/accumulation of IO_3_^−^ was at the same statistical level as after KIO_3_. This may be the consequence of all the transformation of these organic iodine compounds all the way from leaves to grains of the corn; details of the transformation/reduction are not well known.

All tested iodine and vanadium compounds (in the form of ammonium metavanadate) and their combinations had a different effect on the vitamin C content in the grain. The achieved results of corn grain enrichment showed an effective perspective for the biofortification of the generative parts of plants. The testes iodine and vanadium compounds did not have any phytotoxic effect on sweet corn plants as well did not any statistical impact into the yielding.

## Figures and Tables

**Figure 1 molecules-27-01822-f001:**
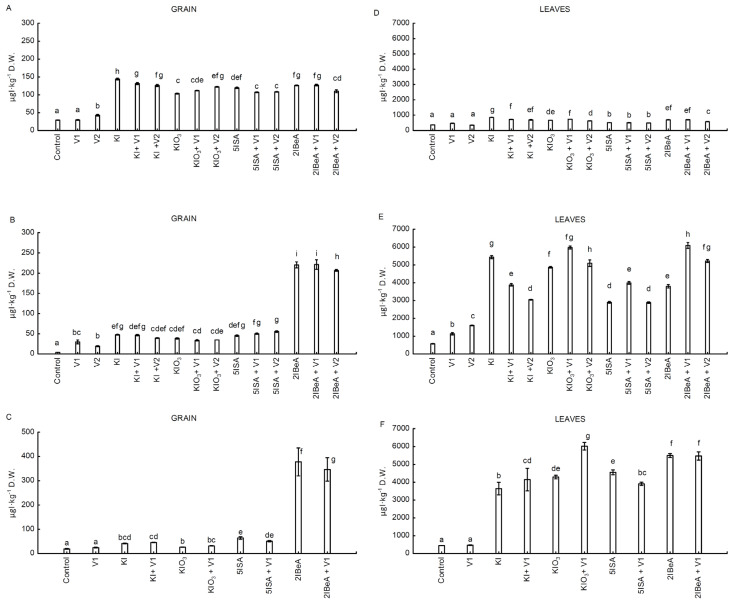
Iodine contents in grain in experiment No. 1 (**A**), No. 2 (**B**) and No. 3 (**C**) and in leaves in experiment No. 1 (**D**), No. 2 (**E**) and No. 3 (**F**) of sweetcorn plants. Means followed by different letters for treatments differ significantly *p* < 0.05 (*n* = 8). Bars indicate standard error.

**Figure 2 molecules-27-01822-f002:**
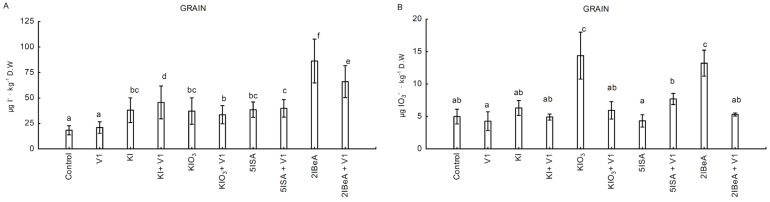
Content of iodide [I^−^] (**A**) iodates [IO_3_^−^] ion (**B**) in sweet corn grain in experiment No. 3. Results only from 1 year of study (2020). Means followed by different letters for treatments differ significantly at *p* < 0.05 (*n* = 8). Bars indicate standard error.

**Figure 3 molecules-27-01822-f003:**
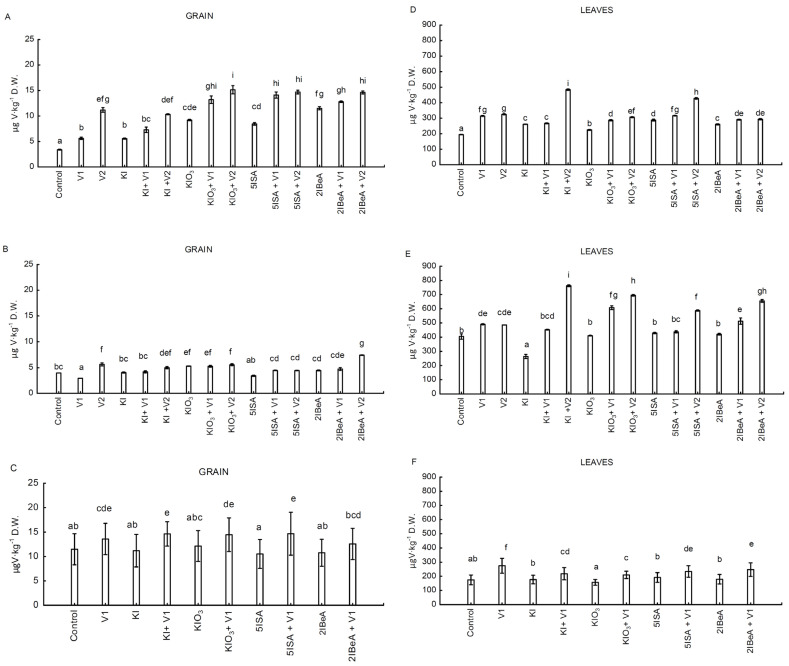
Vanadium contents in grain in experiment No.1 (**A**), No.2 (**B**) and No.3 (**C**) and in leaves in Experiment No.1 (**D**), No.2 (**E**) and No.3 (**F**) of sweetcorn plants. Means followed by different letters for treatments differ significantly at *p* < 0.05 (*n* = 8). Bars indicate standard error.

**Figure 4 molecules-27-01822-f004:**
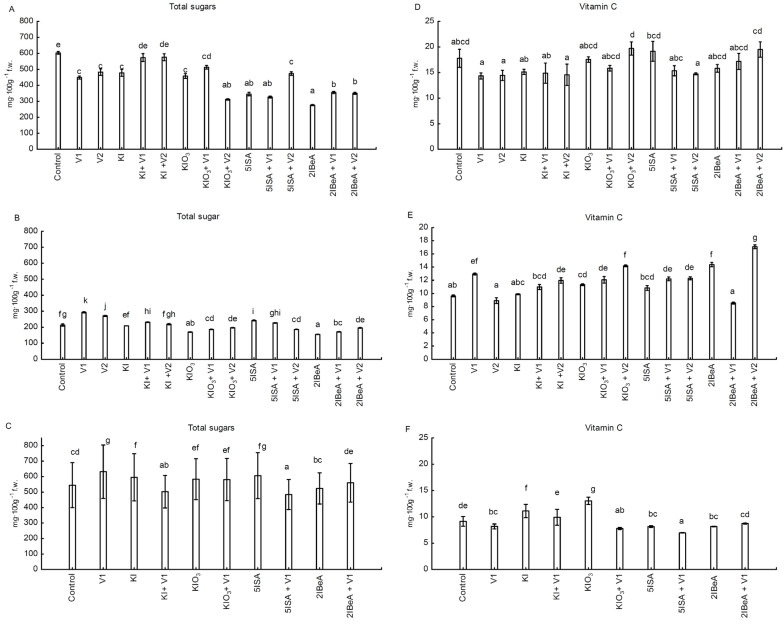
Content of total sugars in grain from experiment No.1 (**A**), No.2 (**B**) and No.3 (**C**), as well as ascorbic acid content in in grain from experiment No.1 (**D**), No.2 (**E**) and No.3 (**F**). Means followed by different letters for treatments differ significantly at *p* < 0.05 (*n* = 8). Bars indicate standard error.

**Table 1 molecules-27-01822-t001:** Results of yield of corn cob, corn cob weight per one plant, corn cob length, number of corn kernels in one row of cob and diameter of one corn cob in experiment No. 1.

Treatment	Yield of Corn Cob (t·ha^−1^)	Corn Cob Weight per One Plant (g)	Corn Cob Length (mm)	Number of Corn in One Rows of Cob	Diameter of One Corn Cob (cm)
Control	10.86 ± 0.91 a	60.73 ± 5.0 a	114.84 ± 3.8 a	9.49 ± 0.3 a	2.94 ± 0.1 a
V1	12.21 ± 1.55 a	64.03 ± 2.9 a	121.18 ± 3.2 a	8.96 ± 0.2 a	2.99 ± 0.1 a
V2	9.78 ± 1.24 a	60.08 ± 6.0 a	107.39 ± 6.0 a	8.49 ± 0.1 a	2.90 ± 0.1 a
KI	7.95 ± 1.29 a	61.25 ± 8.2 a	114.42 ± 9.6 a	9.44 ± 0.1 a	2.95 ± 0.1 a
KI + V1	9.60 ± 2.20 a	57.85 ± 7.5 a	117.48 ± 7.8 a	9.69 ± 0.6 a	2.97 ± 0.1 a
KI + V2	8.39 ± 1.52 a	61.39 ± 7.6 a	120.49 ± 4.4 a	11.72 ± 0.5 a	3.06 ± 0.1 a
KIO_3_	10.50 ± 0.79 a	51.94 ± 22.3 a	119.15 ± 4.7 a	9.91 ± 0.4 a	3.03 ± 0.1 a
KIO_3_ + V1	10.55 ± 0.12 a	55.30 ± 4.1 a	113.98 ± 5.8 a	9.27 ± 0.2 a	3.03 ± 0.1 a
KIO_3_ + V2	12.00 ± 1.85 a	79.52 ± 24.2 a	113.52 ± 7.1 a	11.70 ± 1.4 a	3.05 ± 0.1 a
5ISA	9.55 ± 1.31 a	58.38 ± 2.2 a	114.60 ± 6.9 a	10.77 ± 0.4 a	2.99 ± 0.1 a
5ISA + V1	9.82 ± 0.91 a	61.63 ± 5.6 a	118.59 ± 8.9 a	10.43 ± 0.2 a	3.00 ± 0.1 a
5ISA + V2	9.92 ± 1.70 a	53.50 ± 3.0 a	109.03 ± 3.9 a	10.01 ± 0.3 a	2.89 ± 0.1 a
2IBeA	11.99 ± 1.54 a	58.78 ± 0.9 a	121.76 ± 5.3 a	8.98 ± 0.1 a	3.04 ± 0.1 a
2IBeA + V1	10.10 ± 1.76 a	54.98 ± 2.2 a	114.34 ± 8.5 a	9.43 ± 0.2 a	3.00 ± 0.1 a
2IBeA + V2	10.15 ± 1.75 a	57.68 ± 4.7 a	120.23 ± 11.5 a	9.51 ± 0.3 a	3.06 ± 0.1 a

Means followed by different letters for treatments differ significantly *p* < 0.05 (*n* = 8); ± standard error.

**Table 2 molecules-27-01822-t002:** Results of yield of corn cob, corn cob weight per one plant, corn cob length, number of corn kernels in one row of cob and diameter of one corn cob in experiment No. 2.

Treatment	Yield of Corn Cob (t·ha^−1^)	Corn Cob Weight per One Plant (g)	Corn Cob Length (mm)	Number of Corn in One Rows of Cob	Diameter of One Corn Cob (cm)
Control	11.7 ± 1.29 a	75.73 ± 3.49 a	142.78 ± 3.98 a	7.89 ± 0.11 a	3.19 ± 0.08 a
V1	11.77 ± 3.55 a	71.61 ± 14.10 a	141.09 ± 4.97 a	7.96 ± 0.20 a	3.19 ± 0.08 a
V2	11.48 ± 1.19 a	80.90 ± 4.80 a	147.46 ± 3.94 a	7.81 ± 0.19 a	3.36 ± 0.07 a
KI	12.52 ± 1.99 a	111.31 ± 39.08 a	134.47 ± 6.72 a	7.96 ± 0.04 a	2.93 ± 0.19 a
KI + V1	10.61 ± 1.23 a	65.56 ± 3.90 a	134.87 ± 1.24 a	8.11 ± 0.11 a	2.99 ± 0.06 a
KI + V2	10.58 ± 1.55 a	74.67 ± 11.21 a	134.63 ± 8.52 a	7.81 ± 0.19 a	3.11 ± 0.16 a
KIO_3_	10.41 ± 1.59 a	68.62 ± 4.41 a	139.64 ± 5.20 a	7.94 ± 0.06 a	3.13 ± 0.07 a
KIO_3_ + V1	12.79 ± 2.25 a	68.36 ± 2.79 a	137.33 ± 4.03 a	7.95 ± 0.05 a	3.13 ± 0.04 a
KIO_3_ + V2	12.97 ± 0.83 a	65.42 ± 4.60 a	130.06 ± 7.43 a	7.93 ± 0.08 a	3.03 ± 0.08 a
5ISA	11.70 ± 0.98 a	79.76 ± 7.82 a	141.53 ± 3.49 a	7.88 ± 0.13 a	3.16 ± 0.11 a
5ISA + V1	11.18 ± 1.41 a	68.18 ± 11.23 a	135.14 ± 4.14 a	7.95 ± 0.04 a	3.13 ± 0.14 a
5ISA + V2	11.88 ± 1.22 a	66.77 ± 8.03 a	139.89 ± 3.30 a	7.75 ± 0.25 a	2.95 ± 0.08 a
2IBeA	10.80 ± 1.90 a	69.69 ± 10.75 a	137.60 ± 5.45 a	8.14 ± 0.09 a	3.15 ± 0.14 a
2IBeA + V1	10.67 ± 2.32 a	61.69 ± 8.95 a	136.85 ± 5.94 a	8.17 ± 0.17 a	2.99 ± 0.10 a
2IBeA + V2	9.92 ± 2.45 a	58.83 ± 2.64 a	133.58 ± 6.40 a	7.76 ± 0.14 a	2.92 ± 0.05 a

Means followed by different letters for treatments differ significantly *p* < 0.05 (*n* = 8); ± standard error.

**Table 3 molecules-27-01822-t003:** Results of yield of corn cob, corn cob weight per one plant, corn cob length, number of corn kernels in one row of cob and diameter of one corn cob in experiment No. 3.

Treatments	Yield of Corn Cob (t·ha^−1^)	Corn Cob weight per One Plant (g)	Corn Cob Length (mm)	Number of Corn in One Rows of Cob	Diameter of One Corn Cob (cm)
Control	10.6 ± 0.48 a	69.07 ± 4.03 a	144.04 ± 3.64 a	8.21 ± 0.14 a	3.04 ± 0.07 a
V1	8.9 ± 0.61 a	66.65 ± 5.92 a	136.15 ± 3.73 a	8.09 ± 0.06 a	2.94 ± 0.06 a
KI	9.61 ± 0.45 a	61.10 ± 3.40 a	133.48 ± 3.44 a	8.06 ± 0.06 a	2.97 ± 0.06 a
KI + V1	8.26 ± 0.93 a	63.85 ± 3.52 a	136.30 ± 4.13 a	8.18 ± 0.14 a	3.09 ± 0.05 a
KIO_3_	10.53 ± 0.92 a	63.10 ± 4.73 a	139.29 ± 4.08 a	8.13 ± 0.08 a	3.04 ± 0.06 a
KIO_3_ + V1	11.36 ± 0.91 a	68.94 ± 5.46 a	143.10 ± 4.63 a	7.99 ± 0.10 a	3.08 ± 0.07 a
5ISA	9.6 ± 0.99 a	64.45 ± 3.18 a	141.01 ± 4.56 a	8.10 ± 0.07 a	3.04± 0.06 a
5ISA + V1	10.57 ± 0.81 a	67.21 ± 5.20 a	140.76 ± 4.08 a	8.19 ± 0.21 a	3.09± 0.06 a
2IBeA	9.82 ± 0.63 a	64.70 ± 4.64 a	138.13 ± 3.97 a	8.10 ± 0.15 a	2.98± 0.06 a
2IBeA + V1	11.72 ± 1.02 a	70.87 ± 7.54 a	144.10 ± 3.99 a	8.28 ± 0.22 a	3.09 ± 0.08 a

Means followed by different letters for treatments differ significantly *p* < 0.05 (*n* = 8); ± standard error.

**Table 4 molecules-27-01822-t004:** Content of iodosalicylates [5-iodosalicylic acid (5ISA) and 3,5-diiodosalicylic acid (3,5-diISA)] and iodobenzoates [2-iodobenzoic acid (2IBeA), 2,3,5-triiodobenzoic acid (2,3,5-triIBeA)]—all in µg·kg^−1^ D.W in sweet corn plants grains and leaves.

**Treatment**	**GRAIN µg kg^−1^ D.W**			
	**5ISA**	**3,5-diISA**	**2IBeA**	**2,3,5-triIBeA**
Control	1.65 ± 0.02 a	62.62 ± 0.26 h	2.93 ± 0.29 a	0.200 ± 0.007 ab
V1	52.00 ± 1.38 e	20.36 ± 0.40 b	37.75 ± 0.84 e	0.102 ± 0.004 ab
KI	25.92 ± 1.40 d	24.37 ± 0.11 c	20.85 ± 0.31 d	0.284 ± 0.121 bc
KI +V1	15.73 ± 0.18 bc	28.81 ± 0.34 d	1.80 ± 0.04 a	0.031 ± 0.003 a
KIO_3_	7.92 ± 0.20 ab	17.00 ± 0.46 a	3.95 ± 0.18 a	0.046 ± 0.015 a
KIO_3_ + V1	23.33 ± 0.49 cd	35.16 ± 0.24 f	2.63 ± 0.63 a	0.021 ± 0.001 a
5ISA	60.08 ± 1.38 ef	32.65 ± 0.56 e	13.88 ± 0.37 c	0.159 ± 0.014 ab
5ISA + V1	67.04 ± 5.55 f	38.16 ± 0.24 g	8.81 ± 0.12 b	0.088 ± 0.001 a
2IBeA	2.30 ± 0.10 a	34.42 ± 0.07 ef	365.99 ± 0.83 f	0.465 ± 0.001 c
2IBeA + V1	11.79 ± 0.36 b	39.51 ± 0.73 g	393.61 ± 0.51 g	0.084 ± 0.004 a
	**LEAVES µg kg^−1^ D.W**			
	**5ISA**	**3,5-diISA**	**2IBeA**	**2,3,5-triIBeA**
Control	29.28 ± 0.33 a	15.75 ± 0.09 a	68.61 ± 0.78 d	0.302 ± 0.003 b
V1	59.63 ± 1.50 cd	36.85 ± 0.10 b	30.03 ± 1.33 a	0.6234 ± 0.036 f
KI	48.13 ± 0.23 b	41.08 ± 0.70 c	29.62 ± 0.62 a	0.507 ± 0.048 de
KI + V1	55.75 ± 1.57 c	86.44 ± 0.46 g	29.92 ± 1.06 a	0.099 ± 0.006 a
KIO_3_	60.94 ± 0.34 d	36.32 ± 0.46 b	38.53 ± 0.64 ab	0.618 ± 0.009 f
KIO_3_ + V1	70.14 ± 1.02 e	53.57 ± 1.86 de	52.41 ± 0.11 c	0.397 ± 0.021 c
5ISA	586.37 ± 0.65 f	51.83 ± 0.25 d	43.38 ± 0.60 bc	0.537 ± 0.033 e
5ISA + V1	1082.80± 0.83 g	60.96 ± 0.19 f	26.84 ± 0.20 a	0.461 ± 0.015 cd
2IBeA	55.91 ± 0.67 c	55.76 ± 0.001 e	2552.32 ± 3.86 f	0.250 ± 0.022 b
2IBeA + V1	55.83 ± 1.30 c	42.75 ± 0.09 c	1363.85 ± 6.98 e	0.234 ± 0.011 b

Means followed by the same letters separately for each part of plants are not significantly different for *p* < 0.05; ±, standard error (*n* = 4). Results only from 2020 year of experiment No. 3.

**Table 5 molecules-27-01822-t005:** Percentage of the recommended daily allowance (RDA) for iodine (RDA-I) in a 100 g portion of fresh grains of sweet corn and the hazard quotient (HQ) for intake of I through the consumption of 100 g of fresh grains of sweet corn by adults (70 kg body weight), depending on the foliar application of iodine and vanadium in experiments No. 1, 2 and 3.

	**Experiment No. 1**		**Experiment No. 2**	
**Treatment**	**Percent RDA-I in 100 g of Fresh Sweet Corn Grains (%)**	**HQ-Iodine for 100 g of Fresh Sweet Corn Grains**	**Percent RDA-I in 100 g of Fresh Sweet Corn Grains (%)**	**HQ-Iodine for 100 g of Fresh Sweet Corn Grains** **HQ**
Control	1.039 ± 0.226 a	0.00043 ± 0.00001 a	0.092 ± 0.005 a	0.000063 ± 0.000004 a
V1	0.862 ± 0.121 a	0.00043 ± 0.00002 a	0.979 ± 0.043 c	0.001035 ± 0.000016 d
V2	1.134 ± 0.124 a	0.00063 ± 0.00004 b	0.400 ± 0.034 ab	0.000696 ± 0.000014 c
KI	3.794 ± 0.121 d	0.00212 ± 0.00004 h	0.806 ± 0.035 bc	0.000704 ± 0.000015 c
KI + V1	3.591 ± 0.122 cd	0.00193 ± 0.00005 g	0.867 ± 0.039 c	0.000709 ± 0.000008 c
KI + V2	3.362 ± 0.106 bcd	0.00185 ± 0.00004 fg	0.815 ± 0.028 bc	0.000579 ± 0.000014 b
KIO_3_	2.897 ± 0.086 b	0.00152 ± 0.00002 c	0.792 ± 0.032 bc	0.000549 ± 0.000011 b
KIO_3_ + V1	2.997 ± 0.136 bc	0.00165 ± 0.00001 cde	0.705 ± 0.047 bc	0.000518 ± 0.000008 b
KIO_3_ + V2	3.389 ± 0.099 bcd	0.0018 ± 0.00002 efg	0.660 ± 0.009 bc	0.000515 ± 0.000004 b
5ISA	3.309 ± 0.127 bcd	0.00176 ± 0.00003 def	0.842 ± 0.007 bc	0.000685 ± 0.000014 c
5ISA + V1	2.969 ± 0.048 bc	0.00158 ± 0.00002 c	1.004 ± 0.035 c	0.000715 ± 0.00001 c
5ISA + V2	3.162 ± 0.069 bcd	0.00160 ± 0.00001 c	1.058 ± 0.050 c	0.000767 ± 0.000019 c
2IBeA	3.422 ± 0.123 bcd	0.00186 ± 0.00002 fg	4.618 ± 0.092 e	0.00325 ± 0.000041 f
2IBeA + V1	3.378 ± 0.130 bcd	0.00187 ± 0.00003 fg	4.292 ± 0.298 e	0.003389 ± 0.000048 g
2IBeA + V2	2.938 ± 0.199 bc	0.00162 ± 0.00006 cd	3.751 ± 0.111 d	0.003079 ± 0.000014 e
		**Experiment No. 3**		
**Treatment**	**Percent RDA-I in 100 g of Fresh Sweet Corn Grains (%)**		**HQ-Iodine for 100 g of Fresh Sweet Corn Grains**	
Control	0.387 ± 0.027 a		0.00028 ± 0.00002 a	
V1	0.453 ± 0.022 ab		0.00036 ± 0.00003 b	
KI	0.707 ± 0.026 c		0.00061 ± 0.00001 d	
KI + V1	0.934 ± 0.021 d		0.00067 ± 0.00001 e	
KIO_3_	0.467 ± 0.016 ab		0.00038 ± 0.00001 b	
KIO_3_ + V1	0.604 ± 0.034 bc		0.00046 ± 0.00002 c	
5ISA	1.145 ± 0.111 e		0.00095 ± 0.00009 g	
5ISA + V1	0.987 ± 0.032 de		0.00075 ± 0.00004 f	
2IBeA	6.826 ± 0.860 f		0.00535 ± 0.00078 i	
2IBeA + V1	6.843 ± 0.653 f		0.00527 ± 0.00075 h	

Means followed by the same letters are not significantly different for *p* < 0.05; ±, standard error (*n* = 8). The hazard to consumer exists when the value of HQ exceeds 1.0.

**Table 6 molecules-27-01822-t006:** Design and method of conducting experiments with sweet corn in field experiment; experiments No. 1 (2018 year) and No. 2 (2019 year).

**Treatment**	**Experiment No. 1**					
	**Dose of I Compounds**	**Dose of V as Ammonium Metavanadate**	**Amount of Application**	**BBCH Phases during Foliar Application**	**Intervals between Applications**	**Harvest Phase (BBCH)**
Control	-	-	-	32–61 *	2 weeks	75
V1	-	0.1 µM V	4 times	32–61 *	2 weeks	75
V2	-	1.0 µM V	4 times	32–61 *	2 weeks	75
KI	10 µM I	-	4 times	32–61 *	2 weeks	75
KI + V1	10 µM I	0.1 µM V	4 times	32–61 *	2 weeks	75
KI + V2	10 µM I	1.0 µM V	4 times	32–61 *	2 weeks	75
KIO_3_	10 µM I	-	4 times	32–61 *	2 weeks	75
KIO_3_ + V1	10 µM I	0.1 µM V	4 times	32–61 *	2 weeks	75
KIO_3_ + V2	10 µM I	1.0 µM V	4 times	32–61 *	2 weeks	75
5ISA	10 µM I	-	4 times	32–61 *	2 weeks	75
5ISA + V1	10 µM I	0.1 µM V	4 times	32–61 *	2 weeks	75
5ISA + V2	10 µM I	1.0 µM V	4 times	32–61 *	2 weeks	75
2IBeA	10 µM I	-	4 times	32–61 *	2 weeks	75
2IBeA + V1	10 µM I	0.1 µM V	4 times	32–61 *	2 weeks	75
2IBeA + V2	10 µM I	1.0 µM V	4 times	32–61 *	2 weeks	75
**Treatment**	**Experiment No. 2**					
	**Dose of I Compounds**	**Dose of V as Ammonium Metavanadate**	**Amount of Application**	**BBCH Phases during Foliar Application**	**Intervals between Applications**	**Harvest Phase (BBCH)**
Control		-	-	32–61 *	2 weeks	75
V1		0.1 µM V	4 times	32–61 *	2 weeks	75
V2		1.0 µM V	4 times	32–61 *	2 weeks	75
KI	100 µM I	-	4 times	32–61 *	2 weeks	75
KI + V1	100 µM I	0.1 µM V	4 times	32–61 *	2 weeks	75
KI + V2	100 µM I	1.0 µM V	4 times	32–61 *	2 weeks	75
KIO_3_	100 µM I	-	4 times	32–61 *	2 weeks	75
KIO_3_ + V1	100 µM I	0.1 µM V	4 times	32–61 *	2 weeks	75
KIO_3_ + V2	100 µM I	1.0 µM V	4 times	32–61 *	2 weeks	75
5ISA	100 µM I	-	4 times	32–61 *	2 weeks	75
5ISA + V1	100 µM I	0.1 µM V	4 times	32–61 *	2 weeks	75
5ISA + V2	100 µM I	1.0 µM V	4 times	32–61 *	2 weeks	75
2IBeA	100 µM I	-	4 times	32–61 *	2 weeks	75
2IBeA + V1	100 µM I	0.1 µM V	4 times	32–61 *	2 weeks	75
2IBeA + V2	100 µM I	1.0 µM V	4 times	32–61 *	2 weeks	75

* BBCH 32 [Principal growth stage: Stem elongation- phase 2 nodes detectable]. BBCH 61 [(M) stamens in middle of tassel visible, (F) tip of ear emerging from leaf sheath].

**Table 7 molecules-27-01822-t007:** Design and method of conducting experiments with sweet corn in field experiment No. 3 in 2019 and 2020 years.

Treatment	Experiment No. 3					
	Dose of I Compounds and Dose of I	Dose of V as Ammonium Metavanadate	Amount of Application	BBCH Phases during Foliar Application	Intervals between Applications	Harvest Phase (BBCH)
Control		-	-	61–69 **	3 days	75
V1		0.1 µM V	4 times	61–69 **	3 days	75
KI	100 µM	-	4 times	61–69 **	3 days	75
KI + V1	100 µM	0.1 µM V	4 times	61–69 **	3 days	75
KIO_3_	100 µM	-	4 times	61–69 **	3 days	75
KIO_3_ + V1	100 µM	0.1 µM V	4 times	61–69 **	3 days	75
5ISA	100 µM	-	4 times	61–69 **	3 days	75
5ISA + V1	100 µM	0.1 µM V	4 times	61–69 **	3 days	75
2IBeA	100 µM	-	4 times	61–69 **	3 days	75
2IBeA + V1	100 µM	0.1 µM V	4 times	61–69 **	3 days	75

** BBCH 61 [(M) Stamens in middle of tassel visible, (F) tip of ear emerging from leaf sheath]. BBCH 69 [End of flowering: stigmata completely dry].

**Table 8 molecules-27-01822-t008:** Meteorogical data 2018–2020.

Month	Mean Daily Air Temperature (°C)		
	Year 2018	Year 2019	Year 2020
April	13.3	8.49	6.08
May	16.9	11.44	10.25
June	18.9	20.33	17.09
July	20.6	18.54	18.19
August	20.5	17.27	19.42
September	14.7	13.69	14.47
Mean	17.48	14.96	14.25
	Sum of Rainfall (mm)		
April	29.4	59.1	23
May	59.4	131.7	76.2
June	72.1	14.5	71.8
July	81.7	22.0	55.7
August	52.5	128.7	82.9
September	46.0	78.4	60.0
Sum	341.10	434.4	369.6

## Data Availability

Not applicable.

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
