# Peer review of "Synthesis of Organic Iodine Compounds in Sweetcorn under the Influence of Exogenous Foliar Application of Iodine and Vanadium"

_molecules, 2022, doi:10.3390/molecules27061822_

Round 1
Reviewer 1 Report
The authors describe 3 experiments (201-2019 and 2020), testing iodine application and its synergism with vanadium. They report the findings very descriptive, which is not always needed as the figures contain the required information. In some cases, the manuscript could benefit from giving some more general overview/interpretation of the data. Similarly, the manuscript could benefit from drawing more parallels with literature, previous work or similar studies in other crop. The latter is done in the discussion, but this could occur earlier on. Upon discussion growth of the plant and ascorbate/sugar content, the text could be elaborate to include more on physiological/molecular effects of the interventions. The author could emphasize their general findings more, highlighting the central message or suggestion for further improvements.
See also more specific comments below:
Abstract:
Might contain too much details (exact description of experiments and concentrations).Focus could be more on proper introduction of species used and the current knowledge on iodine/vanadium metabolism there. The detailed description hampers the communications of the main messages, there should be more focus on highlighting the main findings.
Introduction
The statement in line 74-75 regarding the effect of table salt iodization is very strong, and might not be correctly interpreted. Please elaborate (e.g. though iodization has shown a great effect, iodine deficiency often persists), or provide additional references strengthening this point.
The introduction should more extensively discuss the current state of the art in iodine biofortification of corn as well as discuss natural iodine levels in corn. Moreover, occurrence of iodine deficiency in populations depending on corn as a staple could be a part of this analysis, emphasizing the importance of the research. Currently, the introduction does not sufficiently place the research within the current state of the art. This could include information in regard to foliar iodine biofortification in other crops as well.
Results
In the first section (line 125), there is a need to explain the differences within the experiments again. (cf. methods, line 540 onwards). Furthermore, though no significant differences were seen, there could be some commands in regard to witnessed trends or large variations seen.
Figure 1. in some panels, the scale of the y-axis could be changed (particularly panel D), so the differences between the data become more visible (also for figure 3B and F, 4B and 4F).
The link between vanadium/iodine and sugar or vitamin C metabolism should be discussed. The different experiments could benefit from some introduction, stating the incentives. Furthermore, the subdivision of the results ends with 2.3 vanadium content in sweet corn plants, the latter experiments do not fit within this section.
Given the variation in sugar and vitamin C content, would it be possible to elaborate on the effect of total water content, whether or not this is (expected to be ) changed in the samples.
Discussion
The consumption of maize changes significantly from region to region, this could be addressed more in depth (line 306-361), (related to comment in regard to co-occurrence of iodine deficiencies).
The discussion sufficiently includes literature findings, also diving into the underlying molecular mechanisms.
The discussion could go a bit more in depth in regard to the factors which are known to affect vitamin C and sugar content of sweetcorn or other crops more in general. Then, physiological parallels could be drawn towards the physiological effect of the different treatments.
There should be a more elaborate discussion around the percentage of RDA-I displayed. How should this data be interpreted?
The discussion could go more in depth about which application would be the method of choice, taking physiological impact, RDA-I values and cost of application into account.
More specific comments:
Line 17: “rich”
Line 19: “known”
Line 43: “ Iodine of vanadium” seems odd here.
Line 64: consider whether “untested’ is ideally suited here.
Line 68: “0-5 years of age” seems misplaced
Line 70: “idodine”
Line 113: include latin name for plant species
Line 181-182: which two treatments are mentioned here? It seems that a part is missing
Line 207: Something seems incorrect in regard to this sentence.
Line 366: change ”researches” to “research” or “studies”.
Line 376: “the these researches”
Line 433: “proven”
Line 440: “wheat” occurs two times
Line 444: this element (singular)
Line 674: “testes” seems incorrect here
Author Response
Dear Reviewer #1
We would like to thank for your time and efforts put into thorough reading and the review on the manuscript as well as giving us comments and suggestions how to improve our manuscript. We have found all comments valuable and have implemented the corrections according to suggestions. We hope that all the changes and improvements we have introduced into the revised manuscript will meet journal requirements and Editor acceptance. Modified parts in the revised manuscript are marked using the "Track Changes" function in Microsoft Word.

Reviewer 2 Report
molecules-1597905, is an interesting study but lacks with technical and logical flow of information which need to address before consideration for publication.
- Make the title simple and interesting, there is verbosity in the title.
- Abstract is very lengthy and information is moving form Exp1 and Exp 2. Make it more systematic and interesting.
- Introduction contain many sentences with no clear meanings, for instances Line 60-61.
- Remove the short paragraphs and make 2-3 long paragraphs and avoid grammatical and technical mistakes from write up.
- Too much generic information’s given in introduction however, few lines are mentioning the information as per title of the article.
- The objectives of study need more clarity.
- Add the abbreviations full form at the caption of the tables.
- The figures are not very clear, supply high resolution figures.
- Either mention corn or maize throughout the manuscript.
- Prepare sub-headings for the results and describe the results under that sub-heading.
- Remove short paragraphs from the discussion section and prepare 3-4 long paragraphs and remove all those details which is already described in the introduction section of the article.
- Line 503, avoid such mistakes, and make sure all spelling mistakes resolved in the revised manuscript.
- Make subheading in the Material and Method section, and provide with proper citations where necessary.
- Remove short paragraphs from the conclusion and make it short and as per novel results of study coupled with future perspectives.
- Saturate the literature via following recent publications;
Mahmood, A., Wang, X., Ali, A., Awan, MI., Ali, L., Fiaz, S., et al. (2022). Bio-Methane production from Maize with varying nitrogen levels and harvesting times under semi-arid conditions of Pakistan. Pol. J. Environ. Stud. 31 (3):1-9.
Author Response
Dear Reviewer #2
We would like to thank foryour time and efforts put into thorough reading and the review on the manuscript as well as giving us comments and suggestions how to improve our manuscript. We have found all comments valuable and have implemented the corrections according to suggestions. We hope that all the changes and improvements we have introduced into the revised manuscript will meet journal requirements and Editor acceptance. Modified parts in the revised manuscript are marked using the "Track Changes" function in Microsoft Word.

Round 2
Reviewer 2 Report
Article is now in acceptable form however, the figures quality is poor which need to be supplied with high resolution during production.